# The cost of attributing moral blame: Defensiveness and resistance to change when raising awareness to animal suffering in factory farming

Deborah Shulman[1,2], Mor Shnitzer-Akuka[1], Michal Reifen-Tagar[1] *

**1** Baruch Ivcher School of Psychology, IDC Herzliya, Herzliya, Israel, **2** Department of Psychology, Friedrich Schiller University of Jena, Jena, Germany

* michal.reifen@idc.ac.il

**Data Availability Statement:** The data underlying the results presented are available in a public repository (OSF) and can be found at: https://osf.io/eavwt/.

## Abstract

Social change campaigns often entail raising awareness of harm caused by people's behavior. For example, campaigns to reduce meat eating frequently highlight the suffering endured by animals. Such messages may simultaneously attribute moral blame to individuals for causing the harm described. Given people's motivation to protect their moral self-image, we expected that information about the suffering of animals in the meat industry presented with a blaming (versus absolving) frame would generate greater defensiveness and correspondingly resistance to change in support of veg*nism (veganism/vegetarianism). We ran three studies to test this expectation. In two studies, we found that raising awareness of animal suffering using a blaming frame increased defensiveness, leading to lower veg*n-supporting attitudes and behavioral intentions. In one study, our hypothesis was not supported, however, a mini-meta analysis across the three studies suggests the overall pattern is robust. This work expands our understanding of the role of moral self-image preservation in defensiveness and resistance to change, and has applied relevance for the development of effective communication strategies in social and moral campaigns.

## Introduction

Activists for social change often work to raise awareness of the harm caused by current practices or policies, in order to influence people's attitudes. For example, to reduce support for war, anti-war activists often draw attention to the harm innocent civilians experience. Similarly, while trying to increase opposition to abortion, pro-life campaigners often emphasize harm to the fetus. Indeed, researchers have suggested that increasing awareness of harm is a critical first step towards increasing moral concern for a cause and fostering change [1–3]. However, while trying to raise awareness many campaigns simultaneously, and possibly inadvertently, send an additional message of moral blame, which we suggest may undermine their goals.

**Funding:** D.S, M.S & M.R.T Animal Advocacy Research Fund. https://researchfund. animalcharityevaluators.org/ M.R.T Israel Science Foundation (2436/19). Sponsors did not play any role in the study design, data collection and analysis, decision to publish, or preparation of the manuscript.

**Competing interests:** The authors have declared that no competing interests exist.

Animal rights campaigns are a case in point: when groups draw attention to the harm experienced by animals in factory farming, they often do so in a way that directly or implicitly blames meat and dairy eaters for this suffering [4]. For example, in their campaign against eating eggs, animal rights group, Vegan's International Voice for Animals (VIVA!) showed an image of chicks crammed into a blender beside the slogan "the little victims of your morning fry up" [5]. In their video "Chew on it," animal rights group, People for the Ethical Treatment of Animals (PETA) listed reasons for veganism including "because it takes a small person to beat a defenseless animal and an even smaller person to eat one" [6]. Blame can also be implied, rather than overt. For example, the frequently used slogan "meat is murder" aims to sensitize people to the victims of the meat industry by employing a term usually reserved for human targets. The message simultaneously suggests that meat eaters are complicit in murder. Although such messages are meant to raise awareness of the suffering of animals, they also attribute moral blame to the recipient of the message.

Vesting information with moral blame implies a negative evaluation of a person's underlying character [7] and therefore is likely to be threatening their moral self-image. Most people perceive themselves as highly moral even when confronted with information to the contrary [8, 9] and will use a wide range of strategies to maintain their moral self-image [10]. When people are confronted their own immorality, they can experience reduced feelings of self-worth, often leading to self-defensive motivations [11]. Indeed, research in various fields suggests that when people's moral self-image is threatened, they often become defensive [12, 13]. Work specifically on the psychology of meat-eating has found that after imagining being judged by vegetarians, omnivores were more likely to then derogate vegetarians [14] Another study in this domain found that omnivore participants diverted blame for the suffering of animals to mass production, and minimized the role of mass consumption, when asked to discuss their attitudes towards meat eating [15]. Additionally, work on intergroup relations has found that when people's groups are blamed for harming others, they often respond defensively, such as by justifying the social hierarchy from which they are benefitting [16]. Similarly, research focusing on relationships between family members and colleagues, found that the vast majority of people respond to accusations with either denial or justification [17]. In light of this body of research, we suggest that veg*n (vegan or vegetarian) campaigns that introduce information about animal suffering using a blaming frame, risk message recipients feeling morally attacked and becoming defensive and correspondingly resistant to reconsidering their attitudes and behavioral intentions.

People may utilize multiple defense strategies after being blamed for the suffering of animals in the industry. One such strategy is bolstering their belief that humans are superior to animals. Research on the psychology of eating animals, shows that a strong belief that humans are superior to animals helps justify meat consumption [18–20]. Believing that animals are considerably inferior to humans not only justifies past harmdoing towards animals, but also enables people to continue to benefit from industries that harm animals, without regarding it as wrong [21]. Another way that people protect their moral self-image while justifying their behavior is by asserting that the harmdoing in question is not a moral matter [22] and morally disengaging from the issue [19]. People vary in the extent to which they attribute moral worth to animals [18, 23] and many do not view the treatment of animals as a moral issue, but rather consider the practical and material value of animals [24]. Withdrawing moral concern for animals may be an effective way to reduce discomfort regarding meat-eating [25, 26]. If omnivores believe that the harm animals experience in factory farming is not a moral issue, then there is no moral transgression for which they can be held responsible, and therefore their moral self-image can remain intact.

To test our expectation that raising awareness of animal suffering using blaming frames will lead to defensiveness, and in turn, resistance to change, we juxtapose blaming frames with absolving frames that relieve message-recipients of blame. Absolving involves informing people about harm, while reducing their blameworthiness by placing responsibility for harmdoing on an external source. Animal rights groups sometimes utilize this strategy by attributing blame to factory farms instead of consumers, claiming that the factory farms perpetuate and deliberately conceal cruelty from the public [4]. Absolving messages suggest that omnivores are not fully aware of the entailed cruelty, and therefore cannot be held responsible for it. Importantly, absolving does not justify the harm done to animals nor does it suggest that omnivores are less responsible for their *future* harmful behavior; it simply suggests that individuals are not to blame for their *past* involvement in the harmdoing. Compared with blaming frames, absolving messages should be less threatening to people's moral image as they do not entail negative attributions to the self [7]. We therefore hypothesize that recipients who receive information about suffering in an absolving manner may be less defensive, and as such, more open to changing their attitude and behavioral intentions. Even though veg*nism campaigns commonly use both blaming and absolving strategies [4], to the best of our knowledge, the effects of vesting persuasion attempts in these frames have yet to be empirically tested.

## Overview of studies

We conducted three experimental studies to test our hypothesis that introducing information about harm to animals in factory farming using blaming (vs. absolving) frames increases defensiveness, and thereby reduces veg*n supporting attitudes and behavioral intentions. In Study 1, we tested whether being blamed versus being absolved led to increased defensiveness, and reduced support for veg*nism as expressed in both attitudes and behavioral intentions. In Studies 2 and 3, we included an additional control condition with a neutral frame to test whether observed effects were due to the negative effect of being blamed or due to the positive effect of being absolved when presented with information about harm. Finally, we conducted a mini meta-analysis to test the robustness of our findings across our three studies.

In all studies, participants were told that the study was about information processing and memory to reduce demand characteristics. Furthermore, they were informed that they would be presented with an article about one of several topics including the environment, meat-eating, electric bicycles, abortion, or refugees, in order to not only attract participants who were inherently interested in veg*nism. Studies 2 and 3 were preregistered at the Open Science Framework. For each study, the minimal sample size was determined prior to data collection.

All studies were conducted in Israel where there is high meat and dairy consumption [27, 28], like in other developed countries [29]. While the Israeli vegan animal advocacy movement has grown over the last decade [30] and the country has a relatively high number of veg*ns, the vast majority of the population (approximately 87%) are omnivores [31]. Israel currently ranks fourth in meat consumption per capita among OECD countries, behind only the United States, Brazil, and Argentina [28]. The trend in meat consumption per capita in Israel has remained relatively stable over the past decade, and beef consumption has actually increased slightly in the past few years [28]. Altogether, similar to most people in the developed world, most Israelis eat meat and dairy, and in large quantities.

## Study 1

The goal of Study 1 was to test our hypothesis that using blame frames when presenting people with information about harm to animals in factory farming would lead to greater defensiveness and therefore would reduce veg*n supporting attitudes and behavioral intentions. We

tested the mediating role of three defense mechanisms: human superiority beliefs, demoralization, and the 4Ns, which represent common rationalization people use to justify meat-eating. Of note, moralization of veg*nism can also be conceived of as an aspect of attitudes toward veg*nism (as we originally considered). Analyses using this item as part of the positive attitudes measure do not change the pattern or significance of results for this outcome variable, and are included in S1 File. Also, see S1 File for a description of the images included with the text.

## Method

**Participants and procedure.**   The Interdisciplinary Center Herzliya granted ethics approval for the studies. Written consent was obtained for all participants. We recruited 390 Israeli participants via social media. We determined the sample size based on a power analysis for independent sample t-tests using G*Power [32]. It showed that a minimum of 260 participants were required, if this study was to have a power of 0.80, alpha = .05, and a small/medium effect size of $d$ = .35. More participants completed the online study than expected in the time it was available online. Those who did not identify as omnivores were dropped (51) and those who failed an attention check (38) were dropped prior to analysis. This resulted in a final sample of 301 participants (age: $M$ = 30.7, $SD$ = 10, gender: men = 119, women = 182). Participants were randomly assigned to read flyers in Hebrew about the suffering of animals in factory farming. The articles read in the blame and absolve conditions were identical in terms of format and information but differed in the title and opening sentences. In the blame condition, the flyer title was "Eating meat is a violent act" and the opening sentences suggested that readers were aware and therefore responsible for animal suffering (e.g. "It is well known that by consuming meat you support this industry"). In the absolve condition, the title was "What does the meat industry have to hide?" and the opening sentences reflected the idea that the industry conceals information from the public, suggesting that the public is less aware and thus less responsible for animal suffering (e.g. "Most barns and chicken coops are hidden from the public for fear of damaging the industry's profits"). Full manipulation texts can be found in S1 File. Following these opening sentences, all participants were exposed to the same text about the suffering of animals in factory farming. Next, in order to test whether participants across conditions paid equal attention to the information about harm to animals, or alternatively, whether blamed participants engaged less or absorbed less information about animal suffering, participants responded to memory questions assessing the degree of attention they paid to the information in the text. Participants then completed a short survey to measure their defensiveness, attitudes toward veg*nism, and behavioral intentions. At the end of the survey, participants were asked whether they were vegan, vegetarian, or omnivore.

**Measures.**   All items for measures in all studies can be found in S1 File.

*Memory of information*. Participants were asked seven multiple choice questions to test the attention paid to the facts about harm to animals presented in the article (e.g., the natural lifespan of a cow). Participants were given one point for every correct answer, which was then summed to create a memory score for each person.

*Defensiveness*. Defensiveness was measured by three separate variables. First, human superiority beliefs were measured using four items from Dhont and Hodson's human supremacy beliefs scale [18], on a 1(not at all)—7(very much) scale (e.g., *"Animals are inferior to humans,"* $\alpha$ = .71). Second, demoralization of meat consumption was measured with the following item *"To what extent is veganism a moral question, in your opinion?"* on a scale of 1 (not at all)-10 (very much). This item was then reverse-coded to capture demoralization. We used the

16-item 4N scale [33] to measure rationalizations towards meat-eating, asking people to what extent that they thought eating meat is natural, necessary, normal, and nice ($\alpha$ = .93).

*Positive attitudes towards veg\*nism.* Participants answered two questions on a 1 (not at all) -10 (very much) scale ("*To what extent do you support veganism*?", $\alpha$ = .83).

*Positive behavioral intentions towards veg\*nism.* On a 1 (not at all)—10 (very much) scale participants responded to five items (e.g., *"Are you willing to try meatless Monday?" $\alpha$ = .87*).

*Demographics.* Participants reported their gender, age, religiosity, and political ideology.

## Results and discussion

To test our hypotheses, we conducted two-tailed independent sample t-tests, with condition as the independent variable and measures of defensiveness, positive attitudes, and positive behavioral intentions towards veg\*nism as the outcome variables. We then tested whether the effects on attitudes and behavioral intentions were mediated by defensiveness.

We first tested the effects of condition on our hypothesized mediators. Participants in the blame condition scored higher in human superiority beliefs ($M$ = 3.88, $SD$ = 1.37), compared with those in the absolve condition ($M$ = 3.52, $SD$ = 1.38), $t(299)$ = 2.25, $p$ = .025, $d$ = 0.26. Those in the blame condition also demoralized veg\*nism more ($M$ = 5.73, $SD$ = 3.13) than those in the absolve condition ($M$ = 4.78, $SD$ = 2.82), $t(299)$ = 2.75, $p$ = .006, $d$ = .32. There was no significant difference in how much participants rationalized meat-eating using the 4Ns measure (blame condition: ($M$ = 4.17, $SD$ = 1.23), absolve condition ($M$ = 4.07, $SD$ = 1.35), $t(299)$ = .64, $p$ = .521, $d$ = .08).

We then tested the effect of condition on the outcome variables. As expected, found that participants in the blame condition supported veg\*nism less ($M$ = 4.16, $SD$ = 2.26), than those in the absolve condition ($M$ = 5.00, $SD$ = 2.38), $t(299)$ = 3.07 $p$ = .002, $d$ = 0.36. Participants in the blame condition also had less positive behavioral intentions towards veg\*nism ($M$ = 3.77 $SD$ = 2.30), compared with those in the absolve condition ($M$ = 4.65, $SD$ = 2.37), $t(299)$ = 3.24, $p$ = .001, $d$ = 0.38.

**Defensiveness as a mediator of the impact of frames on attitudes and behavioral intentions.** To test whether the defense mechanisms mediated the relationship first, between condition and attitudes, and second, between condition and behavioral intentions, we ran further analysis using Hayes' PROCESS (Model 4) bootstrapping command with 5,000 iterations for SPSS [34]. We conducted parallel mediation analyses to test the mediating role of both human superiority beliefs and the demoralization of veg\*nism. We tested two mediation models—one with positive attitudes towards veg\*nism as the outcome variable, and one with positive behavioral intentions towards veg\*nism as the outcome variable.

*Attitudes.* The mediation analysis revealed that the relative indirect effect of blaming (vs. absolving) on positive attitudes towards veg\*nism via human superiority beliefs (effect = -.12, $SE$ = .06, 95% CI:(-.25, -.01)) and via demoralization (effect = -.47, $SE$ = .18, 95% CI: (-.84, -.13)) was significant (see Fig 1).

*Behavioral intentions.* We then tested whether being blamed (vs. absolved) indirectly influenced behavioral intentions towards veg\*nism through human superiority beliefs and demoralization. Again, the indirect effect of condition via human superiority beliefs (effect = -.17, $SE$ = .08, 95% CI: (-.35, -.02)) and via demoralization (effect = -.30, $SE$ = .12, 95% CI:(-.55, -.08)) was significant (see Fig 2). These results provide support for models in which defensive responses explain the relationship between being blamed (vs. being absolved) and attitudes and behavioral intentions towards veg\*nism.

Finally, we tested whether there was a difference in how much information participants in each condition remembered about the text. There were no significant differences in the extent

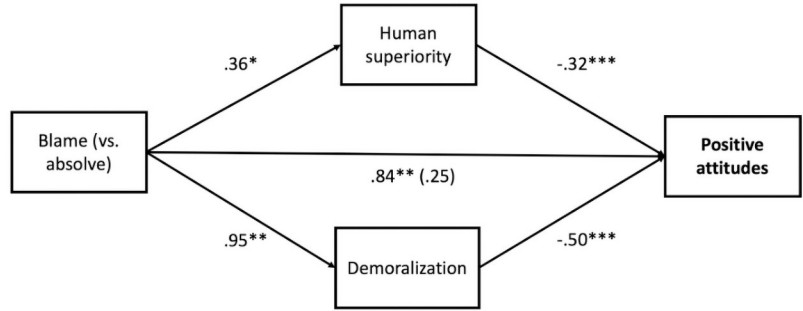

**Fig 1. Model testing the effect of condition (blame vs. absolve) on positive attitudes towards veg*nism through human superiority beliefs and demoralization for Study 1.** *Note.* Unstandardized coefficients displayed. * $p < .05$, ** $p < .01$, ***, $p < .001$, two-tailed tests.

participants remembered information about harm presented between conditions (blame condition: $M = 5.41$, $SD = 1.72$; absolve condition: $M = 5.57$, $SD = 1.48$, $p = .118$, $t(299) = .83$, $p = .409$).

Study 1 provided strong initial support for the hypothesis that blaming is less effective than absolving when trying to persuade people to reduce harmdoing. Specifically, we found that being blamed, versus absolved, when presented with information about harm to animals led to increased defensiveness through human superiority beliefs and demoralization, leading to less positive attitudes and behavioral intentions in support of veg*nism. Surprisingly, we found that being blamed, versus absolved, did not lead to more rationalization of meat-eating in line with the 4Ns. Interestingly, we found no significant difference between the extent participants remembered information about harm to animals between conditions, suggesting that differences in attitudes and behavioral intentions are not due to reduced attention to the information about harm.

## Study 2

The goal of Study 2 was to test the replicability of the findings, and extend our understanding about whether observed differences between conditions in Study 1 were driven by the negative effects of blame or the positive effects of being absolved. Thus, we included a control condition, in which participants were presented with the same information about factory farming, but with a neutral frame. We also added two attention check questions to the study, to check

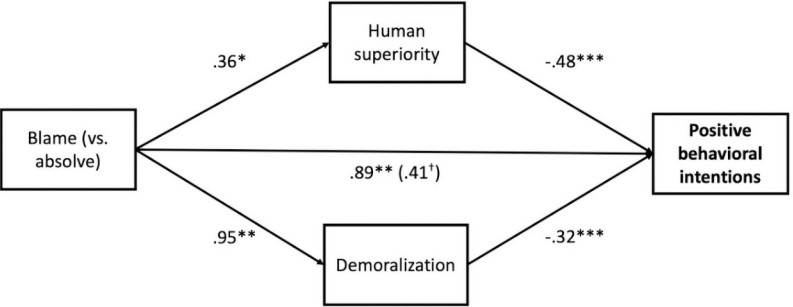

**Fig 2. Model testing the effect of condition (blame vs. absolve) on positive behavioral intentions towards veg*nism through human superiority beliefs and demoralization for Study 1.** *Note.* Unstandardized coefficients displayed. * $p < .05$, ** $p < .01$, ***, $p < .001$, two-tailed tests.

participants were reading the survey questions. This study was preregistered (https://osf.io/u8ezn/?view_only=f54242ef509344baa7b04161348256a2).

Furthermore, we considered that another way that people may respond defensively is by holding weaker efficacy beliefs about their capacity to change their diet. Indeed, research shows that claims of low efficacy, that is, one's perceived inability to change, is a common defense mechanism [35]. In the domain of meat-eating, it has been found that having greater confidence in one's ability to stick to one's diet is positively associated with adherence to a veg*n diet [36]. Moreover, it has been suggested that perceiving less choice in one's ability to reduce meat consumption may be used as a justification for animal-product consumption, while avoiding moral dissonance [37]. Other work has additionally found that individuals with low self-efficacy beliefs about their ability to change their diet are less affected by persuasive message aimed at reducing meat consumption [38]. We, therefore, measured efficacy beliefs as an exploratory variable, and considered that reduced efficacy beliefs may be an additional defense mechanism utilized by those who are blamed. In this study, we did not measure the 4Ns.

## Method

**Participants and procedure.** We recruited 732 participants via an online survey company. As we were able to recruit a large sample via the company, we conducted our power calculation based on the smallest effect size in Study 1, for a one-way ANOVA. This showed that 621 participants were required as a minimum, if this study was to have a power of 0.80, alpha = .05, and effect size of $d$ = .25. To allow for drop out, we aimed to recruit 700 participants, and an extra 32 were collected by the company. Those who failed to recall the name of the article were automatically excluded from participation. We dropped participants who did not identify as omnivore (42) and for failing both attention checks (1). This resulted in a final sample of 689 participants (age: $M$ = 39.16, $SD$ = 13.02, gender: men = 342, women = 347).

The procedure was identical to that of Study 1, except for the additional control condition. The control condition was intended to be neutral. The text was entitled "Consumption of meat in Israel: the facts" and neither blamed nor absolved participants for animal suffering. The opening sentences contained neutral information about factory farming (e.g. "Today, most barns and chicken coops are located in a variety of places in the country") in order to maintain consistent text format across conditions. Following the title and opening sentences, all information presented in the flyer was identical across conditions.

**Measures.** As in Study 1, we measured memory of information, positive attitudes towards veg*nism ($\alpha$ = .88), positive behavioral intentions towards veg*nism ($\alpha$ = .90), participants' defensiveness with separate measures—demoralization and beliefs about human superiority over animals ($\alpha$ = .71), and demographics. We further added a manipulation check to assess the extent to which participants thought that the flyers were blaming and a measure of efficacy beliefs about controlling one's consumption of animal products as an additional defense mechanism.

*Manipulation check.* Participants were asked to what extent they thought the text that they read was blaming, offensive, and respectful (reverse coded), $\alpha$ = .39. As the Cronbach's alpha was very low, we used the item asking participants specifically how blaming they thought the article was as the manipulation check as it most precisely captured the element of blame that we were aiming to manipulate.

*Efficacy beliefs.* Four efficacy items assessed the extent to which participants believed they could take steps towards veg*nism on a scale of 1 (not at all)– 7 (very much) (e.g. "*If I wanted, I could become vegetarian*") $\alpha$ = .76.

## Results and discussion

We conducted planned comparisons to assess differences between conditions. As our hypotheses were directional, we used one-tailed tests (as preregistered) in our analysis. We first checked to see whether the information flyer in the blame condition was seen as more blaming than that in the absolve condition. Surprisingly, we found no significant differences between conditions ($p = 0.22$), suggesting that participants were not significantly affected by the manipulation. We also found no significant differences between conditions for any of variables, and exploratory tests for interactions revealed that none of the demographic variables served as a moderator. As there were no significant differences between conditions, the results of Study 1 were not replicated (descriptive statistics for Study 2 are presented in S1 File).

The different pattern of results between Studies 1 and 2 may be due to differences in sample characteristics. There were difference in age, gender, and ideological orientation between the samples. Those in Study 1 were significantly younger (M = 30.7 years), compared with those in Study 2 (M = 39.2 years). Participants in Study 1 were also mostly women, whereas in Study 2 there were even numbers of women and men. There were also equal numbers of politically liberal and conservative participants in Study 1, while in Study 2 there were twice the number of conservatives. Research findings show that younger people [39], women [40, 41], and liberals [18] are more likely to be veg*n and are more receptive to animal rights campaigns. It should be noted, however, that while there were substantial differences in the study samples, the effect of blame on our outcome variables was not significantly moderated by age, gender, nor political orientation. As well as the differences between sample demographics, participants in this study were recruited by a survey company, received payment, and likely participate in a large volume of studies, unlike those in Study 1, who were recruited on social media. Altogether, it may be that those in Study 2 were inherently less open to dietary change towards veg*nism, paid less attention to the manipulation materials, and were therefore less influenced by them.

## Manipulation check: Pilot study

Given the inconsistency between the strong effects in Study 1 and the null effects in Study 2, we decided to re-run the study with participants recruited via social media like in Study 1. Prior to this, we developed and piloted a revised manipulation check. We aimed to construct a measure with higher reliability and to verify that the blaming text was indeed perceived as more blaming than the absolving text.

A power analysis showed that 128 participants were required if the pilot study was to have a power of 0.80, alpha = .05, and a medium effect size of $d = .5$. We recruited primarily students on social media. To allow for drop out, we recruited 182 participants. After dropping those who did not identify as omnivore (20) and those who failed the attention check (29), 133 participants remained (age = 24.83, $SD = 4.67$, gender: men = 39, women = 94).

Participants were randomly assigned to read the blaming or absolving text. They were then asked to rate the extent to which they felt that the text was "*blaming*"; "*judgmental*"; "*hostile*"; "*derogatory*"; and "*refers to meat-eaters as immoral*" on a scale of 1 (not at all)– 7 (very much). This manipulation check scale had high reliability ($\alpha = .84$). Furthermore, the blaming text was perceived as significantly more blaming ($M = 4.70$, $SD = 1.43$) than the absolving text $M = 3.98$, $SD = 1.40$, $p = .002$, one-tailed test, $d = .51$. This pilot study confirmed that the text frames are meaningfully different in the extent to which they are perceived as blaming.

## Study 3

The goal of Study 3, was to retest our hypothesis. In addition, we added a measure of reactance as an expression of defensiveness. Reactance is experienced when people feel like they are

being told what to do. Reactance generally leads to opposition as people try to assert their autonomy by defying the course of action suggested [42, 43]. We hypothesized that being blamed, versus absolved, would lead to more reactance and therefore would also play a role in leading to less positive attitudes and behavioral intentions in support of veg*nism. This study was preregistered (https://osf.io/rwb5g/?view_only=c74dc5b9370842afb46ae92e1327a2b7).

## Method

**Participants and procedure.** We recruited 389 participants via social media (in different online groups than earlier studies) and on a college campus. Based on the same criteria as in Study 1, a power analysis for a one-way ANOVA showed that 318 participants were required. Those who failed to remember the name of the article of the manipulation text were automatically excluded from participating in the survey. We dropped participants who did not identify as omnivore (71) and participants who failed both attention checks (2). This left us with a final sample size of 316 (age: $M = 30$, $SD = 8.64$, gender: men = 97, women = 219).

The procedure followed the same format as Study 2. Participants were randomly assigned to read about factory farming with either a blaming, absolving, or neutral frame, before answering a short survey.

**Measures.** As in previous studies, we measured memory of information, attitudes towards veg*nism ($\alpha = .88$), and positive behavioral intentions towards veg*nism ($\alpha = .85$), as well as defensiveness with measures of demoralization and human superiority beliefs ($\alpha = .82$); and as in study 2, we included a measure of efficacy beliefs ($\alpha = .79$) and a measure of reactance as exploratory defense mechanisms ($\alpha = .93$). We added one item to the positive attitudes towards veg*nism measure. We also added two items to the human superiority measure, thus using the full scale of Dhont & Hodson (2014) [18].

*Manipulation check*. We included the piloted manipulation check at the end of the study, $\alpha = .89$.

*Reactance*. We adapted items from a Dillard and Shen's reactance measure [44]. Participants rated their agreement with the following statements regarding the flyer on a scale of 1 (not at all)– 7 (very much), (e.g. *"I feel angry that this message tried to pressure me,"* $\alpha = .89$).

## Results and discussion

We conducted planned comparisons between the absolve and blame conditions, and then compared each with the control condition. As in Study 2, since our hypotheses were directional, we used one-tailed tests (as preregistered). While not significant, there was a trend suggesting that the blaming text was seen as more blaming ($M = 4.33$, $SD = 1.71$) than the absolving text ($M = 3.96$, $SD = 1.68$), $t(313) = 1.53$, $p = .063$, $d = .22$. There was also a trend, albeit weaker, suggesting that the blaming text was seen as more blaming than that neutral text in the control condition ($M = 4.01$, $SD = 1.72$), $t(313) = 1.33$, $p = .093$, $d = .19$. There was no difference in how blaming the absolving flyer and the neutral flyer were perceived to be. As in Study 1, there was no significant difference between any of the conditions on memory of information about harm to animals, again suggesting that participants paid similar attention to the texts, across condition.

We then tested the effect of condition on the defense mechanisms. After being blamed, participants were more likely to demoralize veganism ($M = 5.95$, $SD = 2.99$), compared with after being absolved ($M = 5.08$, $SD = 3.17$), $t(313) = 1.95$, $p = .026$, $d = .28$. There was a trend showing that those in the blame condition demoralized veganism more than those in the control condition ($M = 5.32$, $SD = 3.18$), $t(313) =. 1.44$, $p = .077$, $d = .20$. Demoralization did not differ between the absolve and control conditions. Unlike in our first study, there was no significant

difference in human superiority beliefs between any of the conditions. Those in the blame condition had significantly lower efficacy beliefs about their ability to reduce meat and dairy consumption ($M = 4.38$, $SD = 1.46$) compared with those who were absolved of blame ($M = 4.91$, $SD = 1.46$), $t(313) = 2.46$, $p = .007$, $d = .36$. There was a trend showing that those who were blamed had less self-efficacy than those in the control condition ($M = 4.72$, $SD = 1.58$), $t(313) = 1.60$, $p = .055$, $d = .22$. There was no difference between the absolve and control conditions for efficacy. Finally, those who were blamed expressed more reactance against the message ($M = 3.04$, $SD = 1.80$) compared with those who were absolved ($M = 2.54$, $SD = 1.64$), $t(313) = 1.99$, $p = .024$, $d = .29$. There was no difference in reactance between the blame and control condition. There was a trend, however, showing that reactance was lower for those in the absolve condition than those in the control condition ($M = 2.88$, $SD = 1.85$), $t(313) = 1.44$, $p = .076$, $d = .19$.

We next tested the effect of blame on our outcome variables. Those who were blamed had significantly less positive attitudes towards veg*nism ($M = 4.56$, $SD = 2.34$) compared with those who were absolved ($M = 5.22$, $SD = 2.64$), $t(313) = 1.88$, $p = .031$, $d = .26$, replicating the results of Study 1. There was a trend showing that those in the blame condition had less positive attitudes than those in the control condition ($M = 5.03$, $SD = 2.48$), $t(313) = 1.35$, $p = .089$, $d = .02$. There was no significant difference in attitudes between the absolve and control conditions.

Those in the blame condition had significantly less behavioral intentions in support of veg*nism ($M = 3.87$, $SD = 2.20$) compared with those in the absolve condition ($M = 4.67$, $SD = 2.48$), $t(313) = 2.44$, $p = .008$, $d = .34$, again replicating the results of our first study. There was also a significant difference between the blame and control conditions, with those in the control condition expressing more positive behavioral intentions ($M = 4.45$, $SD = 2.26$), $t(313) = 1.78$, $p = .038$, $d = .26$. There was no difference in behavioral intentions between the absolve and control conditions.

**Defensiveness as a mediator of the impact of frames on attitudes and behavioral intentions.** To test whether defense mechanisms mediated the relationship, first, between blame (vs. absolve) and positive attitudes towards veg*nism, and second, between blame (vs. absolve) and behavioral intentions towards veg*nism, we conducted separate mediational analyses using Hayes' PROCESS (2013) bootstrapping command with 5,000 iterations (model 4). As human superiority beliefs were not significantly predicted by condition, this variable was not included in the mediation analyses. To account for the control group, we used the multicategorical function that automatically creates dummy variables, with D1 (absolving vs. blaming) as the independent variable, and D2 (absolving vs. control) as a control.

*Attitudes*. The mediation analysis showed that the indirect effect of condition on positive attitudes towards veg*nism through demoralization [effect = -.44, $SE = .22$, 90% CI (-.82, -.08)] reactance [effect = -.14, SE = .07, 90% CI (-.27, -.02)], and efficacy [effect = -.17, $SE = .08$, 90% CI (-.31, -.06)] were all significant (see Fig 3).

*Behavioral intentions*. The mediation analysis revealed that the indirect effect of condition on behavioral intentions towards veg*nism through demoralization (effect = -.32, $SE = .16$, 90% CI (-.58, -.06)), reactance (effect = -.17, $SE = .08$, 90% CI (-.32 -.03)), and efficacy (effect = -.15, $SE = .07$, 90% CI (-.28, -.04)) were all significant (see Fig 4). These results again suggest that defensiveness may explain the effect of the blaming texts on resistance to changing attitudes and behavioral intentions.

Overall, the results of Study 3 were largely consistent with those of Study 1. We found that compared with being absolved, being blamed for the suffering of animals, led to more defensiveness and less positives attitudes and behavioral intentions towards veg*nism. There were several defense mechanisms that mediated these effects namely, demoralization, reduced

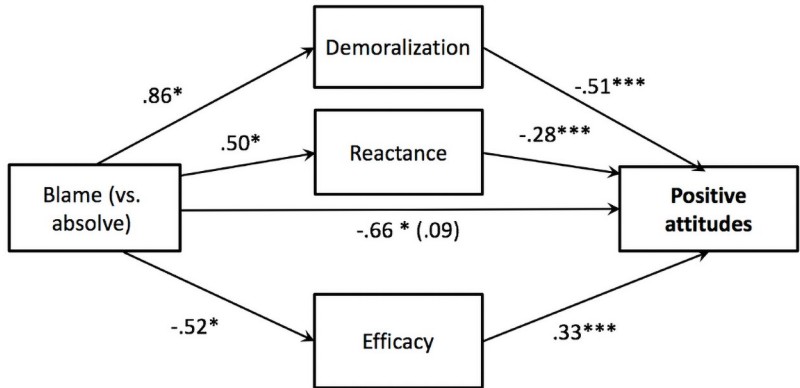

**Fig 3. Model testing the effect of condition (blame vs. absolve) on positive attitudes towards veg\*nism through demoralization, reactance, and efficacy beliefs for Study 3.** *Note.* Unstandardized coefficients displayed. $^*p < .05$, $^{**}p < .01$, $^{***}p < .001$, one-tailed tests.

efficacy beliefs, and reactance, though not human superiority. In other words, those who were blamed (vs. absolved) regarded veg\*nism as *less* of a moral issue, believed that they were *less* able to change their diet towards veg\*nism, and felt that the information that they read about animal suffering (with the blame-frame) was a threat to their freedom of choice. Of note, the manipulation check was below the threshold for significance ($p = .063$) unlike in the manipulation check pilot study in which it was highly significant ($p = .002$). This could be because it was measured after participants had already responded to the defense mechanism measures, which might have alleviated their sense of blame, or because it was measured at the end of the survey and thus more removed from the text, whereas in the pilot study it was measured directly after participants read the text.

In this study, we added a control condition to better understand whether the differences between conditions were due to being absolved or being blamed. The pattern of results suggests blame may play a larger role than being absolved in driving the effects on attitudes and behavioral intentions via defensiveness. This is because there were differences, significant and

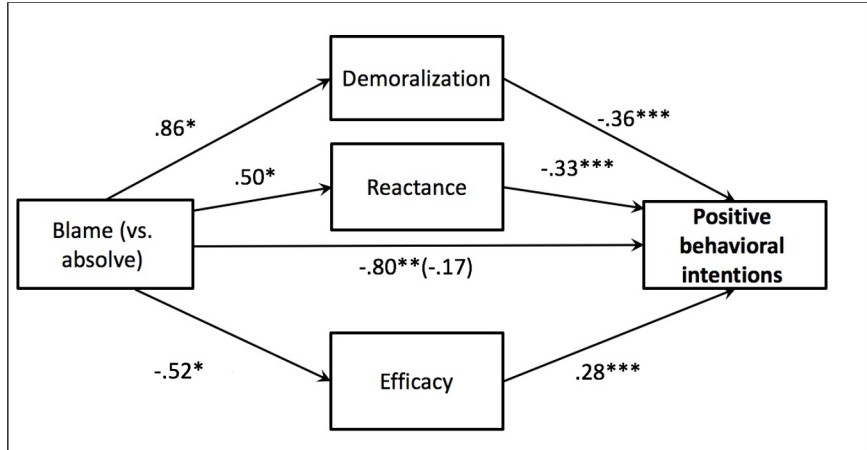

**Fig 4. Model testing the effect of condition (blaming vs. absolving) on positive behavioral intentions towards veg\*nism through demoralization, reactance, and efficacy beliefs for Study 3.** *Note.* Unstandardized coefficients displayed. $^*p < .05$, $^{**}p < .01$, $^{***}$, $p < .001$, one-tailed tests.

trending, between the blame and control condition for most variables. The only significant difference between the absolve and control condition was for reactance—those who were absolved expressed less reactance than those in the control condition. Although the overall pattern suggests that it is being blamed that leads to greater defensiveness and less change in attitudes and behavioral intentions, the largest effects are consistently between the blame and absolve conditions, suggesting that both blaming and absolving play some role.

### Mini meta-analysis of main effects across studies

To assess the overall effect of blaming (vs. absolving) on attitudes and behavioral intentions towards veg*nism across the three studies, we conducted a mini-meta analysis [45]. Overall, the main effects across studies were significant. Those in the blame condition had less positive attitudes towards veg*nism than those in the absolve condition (M $d$ = .20, $Z$ = 3.01, 95% CI [0.07, 0.32], $p$ = .002, two-tailed). Those in the blame condition had less positive behavioral intentions towards veg*nism than those in the absolve condition (M $d$ = .18, $Z$ = 2.73, 95% CI [0.05, 0.31], $p$ = .006, two-tailed). The overall effects are fairly small but clearly significant and suggest that introducing information about harm to animals using blaming (vs. absolving) frames leads to less positive attitudes and behavioral intentions towards veg*nism.

## General discussion

In this work, we hypothesized that presenting information about harm to animals in factory farming using a blaming (versus absolving) frame would lead to increased defensiveness, and in turn, resistance to change in both attitudes and behavioral intentions. To test this, we ran three studies in which omnivore participants were presented with information that detailed the harm experienced by animals in factory farms, preceded either by a blaming or an absolving title and opening. Following this, participants responded to measures of defensiveness, which could potentially serve to protect their moral self-image, and reported their attitudes and behavioral intentions towards veg*nism.

The results of a mini-meta analysis combining the blame and absolve conditions across studies ($N$ = 961) showed that the blaming frame (vs. absolving frame) led participants to support veg*nism less, expressed in both attitudes and behavioral intentions. We found strong support for our hypothesis in Study 1. Specifically, being blamed (vs. absolved) led to less support for veg*nism, via an increase in two distinct defense mechanisms: human superiority and demoralization. In Study 2, we did not replicate these results. In Study 3, we found that being blamed (vs. absolved) led to reduced positive attitudes and behavioral intentions towards veg*nism, like in Study 1, and that these effects were mediated through the increased use of defense mechanisms: again demoralization, reactance, and reduced efficacy beliefs, but not human superiority beliefs. Across the studies, we also tested and found no difference in the extent to which participants recalled the information about harm across conditions. This suggests that the effect of being blame (vs. absolved) on resistance to attitudinal and behavioral change was not driven by differential recollection of the information presented, but rather, was due to the defensive ways in which participants who were blamed contended with it. To the best of our knowledge, this is the first demonstration of the costly effects of blame in persuasion efforts, and specifically with regards to raising awareness of animal suffering in factory farming.

From an applied perspective, our results suggest that activists and campaigners should seriously consider that people care deeply about their moral image. Attributing blame may be an instinctive response for those who are invested in moral issues (also beyond veg*nism). However, activists would do well to impart powerful and thought-provoking information while

maintaining the integrity of their target audience's moral self-image, in order not to elicit defensive reactions. If information is provided in a way that can be construed as blaming, self-protective and justifying responses may follow and campaigns may even backfire.

The pattern of findings across the studies highlights the importance of exploring boundary conditions for our hypothesis. In Studies 1 and 3, where we found that blaming (vs. absolving) led to less positive attitudes and behavioral intentions towards veg*nism, participants were recruited mainly via social media and college campuses and were primarily students; whereas in Study 2, where we found no effect of blaming, participants were paid through an online survey company. As well as being paid, Study 2 participants were significantly older, and the sample consisted of relatively more men and more political conservatives, than in Studies 1 and 3. A wealth of research suggests that younger (vs. older) people [46], women (vs. men) [39], and liberals (vs. conservatives) [18, 23] are more likely to be veg*n and are more open to veg*nism, suggesting that they may be more receptive to veg*nism campaigns at the outset. In Israel too, veganism is positively associated with being younger, being a woman, and being liberal, and this dietary choice may even be seen as incompatible with opposing social identities [30, 47]. It could thus be that participant in Studies 1 and 3, were more favorable to veg*nism, compared with those in Study 2. While age, gender, and political orientation did not moderate the effects of blaming (vs. absolving) on attitudes and behavioral intentions towards veg*nism in the current studies, a question for future research is whether other related variables may moderate the effect.

Defensive responses might be more or less pronounced among those with affinities to different social identities or cultures. Existing work suggests that there are multiple factors that can influence meat-eating behavior, including values, beliefs, emotions, or external incentives such as price and availability, and that these may vary for groups of people with particular identities [29]. In the real world, veg*n animal advocacy groups not only make moral arguments for veg*nism, they also employ other strategies in a bid to reduce animal-product consumption, such as promoting personal health messages, promoting flexible diets, and drawing on social norms. It could be that for those who might be more inclined towards veg*nism for ethical reasons at the outset, moral messages, like the ones we tested in our study, may have more influence. However, for other groups of people who are more motivated to preserve their health, for example, older populations, health messages may be more effective [29]. This is a question that future work could explore.

Across our studies, we measured several different possible defense mechanisms. Overall, defensiveness mediated the relationship between blaming (vs. absolving) and attitudes and behavioral intentions, but further research is merited to examine which are the more meaningful defense mechanisms and for whom. Demoralization was a consistent mediator across the two studies in which main effects were found (Study 1 and 3) whereas human superiority beliefs mediated the effects in Study 1 but not 3. Furthermore, both efficacy beliefs and reactance were meaningful mechanism in Study 3, but were not examined in Study 1. This suggests that there are multiple defensive strategies that lead to resistance to change, which is in line with previous research demonstrating the many ways that people are defensive in the domain of meat-eating (e.g. [19, 37, 48]). Future research should investigate which defense mechanisms are more prevalent among different groups of individuals, situations, and cultural contexts.

Another interesting question that is beyond the scope of this paper is whether assigning blame at the group level (e.g., society, omnivores) would be more effective than blaming the individual for their harmdoing. While one could argue that the threat associated with blame may be diffused if targeted at a collective, we might expect that because one's identity and self-esteem is generally strongly intertwined with one's group identity [49], blame directed at one's

group may be equally counter-productive. Finally, examining the long-term effects of being blamed is important for both theoretical and applied purposes. More time to consider a blaming message may simply provide people with greater opportunity to further justify their opinions and could thus entrench their existing attitudes even more. However, given that our work looked at the immediate response to blaming messages, more research is necessary to test this empirically.

## Conclusion

Limitations notwithstanding, our findings contribute to the literatures on social change, moral self-image maintenance and to the literature on the psychology of veg*nism. The results suggest that the way in which activists communicate ethical transgressions impacts their effectiveness. Activists in social movements, who are often motivated by strong moral convictions about their cause [50], might be inclined to blame people, either intentionally or inadvertently, for their complicity in unethical behavior, as they try to raise awareness and promote their moral agenda. Our work warns of the unintended negative consequences of blaming and suggests that blame leads to increased defensiveness and may be counter-productive. This work indicates that it may be more effective to alleviate people of responsibility for their past harm-doing while informing them about the harm in which they are involved, in order to increase receptiveness to information about harm and consequently bring about change.

## Supporting information

**S1 File. Supporting information file.**
(DOCX)

## Acknowledgments

We thank Eran Halperin for his comments on early drafts of this manuscript. We also thank Tamar Gur for her suggestions as we were designing Study 2, and Yuval Grinstein for help with data collection.

## Author Contributions

**Conceptualization:** Deborah Shulman, Mor Shnitzer-Akuka, Michal Reifen-Tagar.

**Formal analysis:** Deborah Shulman.

**Funding acquisition:** Deborah Shulman, Mor Shnitzer-Akuka, Michal Reifen-Tagar.

**Investigation:** Deborah Shulman, Mor Shnitzer-Akuka.

**Methodology:** Deborah Shulman, Mor Shnitzer-Akuka, Michal Reifen-Tagar.

**Supervision:** Michal Reifen-Tagar.

**Writing – original draft:** Deborah Shulman, Mor Shnitzer-Akuka.

**Writing – review & editing:** Michal Reifen-Tagar.

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
