## [Decision Letter · Decision Letter 0]

14 May 2021

PONE-D-20-39949

The Cost of Implying Blame when Raising Awareness to Animal Suffering in Factory Farming: Defensiveness and Resistance to Change

PLOS ONE

Dear Dr. Shulman,

Thank you for submitting your manuscript to PLOS ONE. After careful consideration, we feel that it has merit but does not fully meet PLOS ONE’s publication criteria as it currently stands. Therefore, we invite you to submit a revised version of the manuscript that addresses the points raised during the review process.

Indeed, I have read with great pleasure your manuscript, which is interesting, deals with an important topic, is well written and engaging. The transparency of the research, i.e., the fact that two of the studies are pre-registered, with data available in an open archive, is definitely a plus. It would have been even better if the data had been made available for the review process, which I would suggest for the next step of the revision process, for the sake of full transparency.

I would like to express my thanks to the two reviewers for their precious work. As you will see, they share with me the positive assessment and provide interesting suggestions to improve the manuscript. Indeed, they pointed to important aspects that should be elaborated upon, in the introduction and in the discussion, and provided useful references.

All the points raised by the reviewers deserve careful attention and should be addressed in your rebuttal letter.

Among other things, I agree with the observation that there are terms in the manuscript that might be obscure to some of the larger audience, hence brief explanations might be useful, and that it would be appropriate to provide some more information about the most widespread dietary habits in Israel, which you write to be characterized by the fact that the ‘animal product consumption is common' (page 5). First (but this is a very minor point), it is not clear whether (page 5) if it is meant here that in Israel most people are omnivore, or that they are not vegan. In other words, here with ‘animal product’ are you referring to ‘meat’ or also eggs and milk products? More importantly, regarding meat and animal product consumption in Israel, are these similar or different from those of highest meat consumption countries (e.g., US, Argentina, Australia)? Is the consumption of meat and animal products increasing, decreasing, or stable in Israel? Especially considering the question of generalizability of your results to other countries, this is important information for the reader.

The difference between the results of study 2a, and those of the other two studies is very interesting and it may be theoretically relevant. Self-selection may be at stake in studies 1 and 2b, meaning that those people who are already interested in veg* eating styles are more likely to be affected by the message and/or its characteristics, and this should be addressed in the limitations. 

Finally, I have some methodological notes:

More information should be provided for the power analysis (in all studies). E.g., Study 1:  “a minimum of 260 participants were required, if this study was to have a power of 0.80, alpha = .05, and a small/medium effect size of d = .35.” 

Please clearly state for what test this number of participants is adequate: Is it for an independent samples t-test? or for the mediation analysis? Furthermore, when chosing the alpha level, how was the number of tests considered (e.g., through Bonferroni correction?)

Minor points: 

I found the wording “independent T-tests” (e.g., Page 8, “results and discussion” section) a bit confusing. I suggest to write: “independent sample t-tests” (lowercase)Page 9, also the following sentence was a bit confusing: “model 4 was used to test parallel mediation).” What do you mean exactly? Does it mean that you run two separate analyses, one for each dependent variable?Study 1, the mediation analyses would be easier to follow if you could present them graphically, in a similar way to what you did for study 2b

Besides, I wonder why you chose to name them Study 1 - 2a - 2b, instead of Study 1 - 2 -3, especially because 2a and 2b are not identical to each other, so I found this a bit confusing (but this may be a matter of personal preferences).

We look forward to receiving your revised manuscript.

Kind regards,

Cristina Zogmaister, Ph.D.

Academic Editor

PLOS ONE

Journal Requirements:

Reviewers' comments:

Reviewer's Responses to Questions

**Comments to the Author**

1. Is the manuscript technically sound, and do the data support the conclusions?

Reviewer #1: Yes

Reviewer #2: Yes

2. Has the statistical analysis been performed appropriately and rigorously? 

Reviewer #1: Yes

Reviewer #2: I Don't Know

3. Have the authors made all data underlying the findings in their manuscript fully available?

Reviewer #1: Yes

Reviewer #2: Yes

4. Is the manuscript presented in an intelligible fashion and written in standard English?

Reviewer #1: Yes

Reviewer #2: Yes

5. Review Comments to the Author

Reviewer #1: Thank you for including me in this process. I thoroughly enjoyed the paper and think that it is a valuable contribution to the field.

The main issue I have with the manuscript is the exclusion of many relevant studies in the introduction, and especially in the discussion. I have suggested a few (there are more) at the end of the review. This literature will help unpack and discuss results in relation to possible contextual features, moralisation/demoralisation, efficacy etc.

I was also expecting a discussion on identities/values attached to groups, such as the issues with merging all non-meat eaters even though you are focusing on omnivores.

I think it would be useful to expand on the discussion about the Israeli diet context you touch upon that there is a need to test this in further contexts (p.27), but it would be useful for the reader to know more about the Israeli context and why there might be contextual differences (or not) – see for example Schwartz, 2020).

The context could also feed into the moralisation/demoralisation in terms of group belonging/cultural belonging evoked by the different condition texts – you almost touch upon this on p.27 (“but further research is merited to examine which are the more meaningful defense mechanisms and for whom.”) but I think you could elaborate with a sentence or two.

The efficacy discussion could be related to a lot of the diet sustainability literature.

Finally, I am missing a more direct discussion of the practical implications of your research, you have a brief description of what to do and not, but this could be elaborated more on in terms of value for campaigns, social movements etc. (even beyond the veganism) to emphasise the usability.

Some minor (grammatical) points:

Inconsistent use of v*gn and veg*n

(p.20) “Furthermore, those the blaming text was perceived as significantly more blaming” – missing a ‘reading/assigned’

Some suggested references:

Amiot, C. E., & Bastian, B. (2017) Solidarity with animals: Assessing a relevant dimension of social identification with animals. PLoS ONE, 12(1): e0168184. https://doi.org/10.1371/journal.pone.0168184

Becker, J. C., Radke, H. R. M., & Kutlaca, M. (2019). Stopping wolves in the wild and legitimizing meat consumption: Effects of right-wing authoritarianism and social dominance on animal-related behaviors. Group Processes & Intergroup Relations, 22(6), 804–817. https://doi.org/10.1177/1368430218824409

Caviola, L., Everett, J., & Faber, N. S. (2019). The moral standing of animals: Towards a psychology of speciesism. Journal of Personality and Social Psychology, 116(6), 1011–1029. https://doi.org/10.1037/pspp0000182

Cruwys, T., Norwood, R., Chachay, V. S., Ntontis, E., & Sheffield, J. (2020). "An important part of who I am": The predictors of dietary adherence among weight-loss, vegetarian, vegan, paleo, and gluten-free dietary groups. Nutrients, 12(4), 970. https://doi.org/10.3390/nu12040970

Leach, S., Sutton, R. M., Dhont, K., & Douglas, K. M. (2020). When is it wrong to eat animals? The relevance of different animal traits and behaviors. European Journal of Social Psychology. https://doi.org/10.1002/ejsp.2718

Leite, A. C., Dhont, K., & Hodson, G. (2019). Longitudinal effects of human supremacy beliefs and vegetariaism threat on moral exclusion (vs. inclusion) of animals. European Journal of Social Psychology, 49(1), 179-189. https://doi.org/10.1002/ejsp.2497

Finally, I would again like to thank the editor and authors for including me in this process. I think the paper is very interesting and a valuable contribution to the field.

Reviewer #2: Regarding the literature review: You do a good job of showing the practical problems with using a blame frame (efficacy issues in having it backfire psychologically resulting in rejection), but I think you should also mention what benefits may come from using a frame that puts some responsibility on the consumers themselves who finance this industry (or them as citizens who elect political leaders that endorse these farming laws and practices). If the blame is warranted, is there something valuable in admitting a consumer’s culpability in the problem while also showcasing how they are then a part of the solution? Or is the ultimate test of whether a campaign is “successful” based on immediate responses to messages and short term behavior change? Might there be long-term or deeper benefits to assigning a blame frame that an experiment cannot measure? (to add to your limitations section). This could be acknowledged.

Your own study’s findings in the end conclude “This work indicates that it

may be more effective to alleviate people of responsibility for their past harmdoing while

informing them about the harm in which they are involved, in order to increase receptiveness

to information about harm and consequently bring about change.”

A question to consider (although it was not tested in the study but could be acknowledged as an area for future research): Could the blame be allocated more collectively/socially (our society/culture is problematic because it validates and socially sanctions this mass consumption of animals) rather than aimed directly at them as one individual consumer? And would this not reduce the defensiveness and perhaps engage people in terms of citizenship (not just their role as consumer).

METHOD: Decent sample size of 300+ study 1 –recruited via social media (and had more women). Doubled for study 2a bc paid participants from a hired survey company. Good that you noted the differences in sample sourcing may affect results. Study 2b (316 participants, many more women). Plus a mini meta analysis. While I am not a quantitative researcher, from what I could assess, I found the three messages (blaming, absolving, and neutral) to be appropriate as a testing tool, as well as the measures for other categories such as 4Ns, human superiority, dietary change efficacy, etc. (in reading your supplementary materials). I could tell you as researchers understand vegan advocacy and animal rights issues.

I like that in study 2a you considered contravening variables and “therefore, measured efficacy beliefs as an exploratory variable, and considered that reduced efficacy beliefs may be an additional defense mechanism utilized by those who are blamed.” In general, I find that the researchers were self aware in acknowledging weaknesses/limitations and potential confounding variable.

I also like that you ran an extra test with study group one to test some inconsistencies from Study 2a: “Given the inconsistency between the strong effects in Study 1 and the null effects in

Study 2a, we decided to re-run the study with participants recruited via social media like in

Study 1.” (133 participants: many more women)

I think you should explain further what is meant by this finding in study 2b using terms that activists could understand. “There were several defense mechanisms that mediated these effect namely,

demoralization, reduced efficacy beliefs, and reactance, though not human superiority.” What implication does this have as far as providing tangible guidance for campaign designers? (add that to the conclusion).

From the conclusion:

This is quite significant: “To the best of our knowledge, this is the first

demonstration of the costly effects of blame in persuasion efforts, and specifically with

regards to raising awareness of animal suffering in factory farming.”

A question that arose for me after reading study 1: Are you saying that being blamed for farmed animal suffering won’t make you rationalize meat eating as much as it will make you defend your human superiority over other animals? (although either as well as both would be detrimental to dietary behavior change). This would be an area for more granular/nuanced study in what messages may be effective here in seeing if it is human superiority beliefs that most need changing among omnivores rather than focusing on changing beliefs about meat eating. Or perhaps you can think of more specific ways this finding is significant for vegan advocacy campaign designers and could be further examined by scholars.

Typo pg 18: make “effect” plural in the middle sentence “There were several defense mechanisms that mediated these effect namely”

Important study. Useful. Well done.

6. PLOS authors have the option to publish the peer review history of their article (what does this mean?). If published, this will include your full peer review and any attached files.

Reviewer #1: **Yes: **Sara Vestergren

Reviewer #2: **Yes: **Carrie P. Freeman

---

## [Author Response · Author response to Decision Letter 0]

9 Jun 2021

1 June, 2021

Dr. Zogmaister

Academic Editor

PLOS ONE

Dear Dr. Zogmaister,

We are very glad for the opportunity to refine our manuscript and resubmit it to PLOSONE following revisions. 

Thank you very much for all the time and thought you and the reviewers put

into giving us such constructive feedback. We greatly appreciate it and believe that it has considerably strengthened the paper.

In revising the manuscript, we carefully considered all the concerns and suggestions raised. Below, we describe the changes that we have made in response to each of the points. 

Sincerely,

Deborah Shulman

Mor Shnitzer-Akuka

Michal Reifen-Tagar, Ph.D.

 

Editor’s comments 

1.1. General feedback

“I have read with great pleasure your manuscript, which is interesting, deals with an important topic, is well written and engaging. The transparency of the research, i.e., the fact that two of the studies are pre-registered, with data available in an open archive, is definitely a plus. It would have been even better if the data had been made available for the review process, which I would suggest for the next step of the revision process, for the sake of full transparency. I would like to express my thanks to the two reviewers for their precious work. As you will see, they share with me the positive assessment and provide interesting suggestions to improve the manuscript. Indeed, they pointed to important aspects that should be elaborated upon, in the introduction and in the discussion, and provided useful references.”

Thank you for such encouraging feedback. We were very excited to work on the manuscript after reading all the comments and suggestions, and believe that we have managed to address them in the revised manuscript. 

We have also now uploaded the data for all studies to the OSF website for full transparency and included this link on p.6 on the paper: https://osf.io/d2jfb/?view_only=82578dec80154aa994826c36c652f324

1.2. Including more information about dietary habits in Israel

“It would be appropriate to provide some more information about the most widespread dietary habits in Israel, which you write to be characterized by the fact that the ‘animal product consumption is common' (page 5). First (but this is a very minor point), it is not clear whether (page 5) if it is meant here that in Israel most people are omnivore, or that they are not vegan. In other words, here with ‘animal product’ are you referring to ‘meat’ or also eggs and milk products? More importantly, regarding meat and animal product consumption in Israel, are these similar or different from those of highest meat consumption countries (e.g., US, Argentina, Australia)? Is the consumption of meat and animal products increasing, decreasing, or stable in Israel? Especially considering the question of generalizability of your results to other countries, this is important information for the reader.”

Thank you for this suggestion. We have addressed it by adding in key information about dietary habits in Israel in the introduction (p. 6):

“All studies were conducted in Israel where there is high meat and dairy consumption (Ministry of Agriculture & Rural Development, 2018; OECD, 2020), like in other developed countries (Stoll-Kleemann & Schmidt, 2017). While the Israeli vegan animal advocacy movement has grown over the last decade (Gvion, 2020; Shwartz, 2020) and the country has a relatively high number of veg*ns, the vast majority of the population (approximately 87%) are omnivores (Cohen, 2015). Israel currently ranks fourth in meat consumption per capita among OECD countries, behind only the United States, Brazil, and Argentina (OECD, 2020). The trend in meat consumption per capita in Israel has remained relatively stable over the past decade, and beef consumption has actually increased slightly in the past few years (OECD, 2020). Altogether, similar to most people in the developed world, most Israelis eat meat and dairy, and in large quantities.”

In the discussion section, we also mention that the social identity of veg*ns in Israel is similar to that of veg*ns in other researched contexts (p. 23)

“In Israel too, veganism is positively associated with being younger, being a woman, and being liberal, and this dietary choice may even be seen as incompatible with opposing social identities (Gvion, 2020; Shwartz, 2020).”

1.3. Self-selection 

The difference between the results of study 2a, and those of the other two studies is very interesting and it may be theoretically relevant. Self-selection may be at stake in studies 1 and 2b, meaning that those people who are already interested in veg* eating styles are more likely to be affected by the message and/or its characteristics, and this should be addressed in the limitations.

Thank you for this comment. We actually shared this concern at the outset, and as such, in all studies we used a cover story and told participants that the study was about information processing and memory and that they would be presented with an article about one of several topics including the environment, meat-eating, electric bicycles, abortion, or refugees. We did this in order to not only attract participants who were inherently interested in veg*nism. We assumed that we noted this in the manuscript, but realize now that we had not provided full details, so thank you for drawing our attention to this. We have now added this information on p.5.

1.4. Sample size calculation

More information should be provided for the power analysis (in all studies). E.g., Study 1: “a minimum of 260 participants were required, if this study was to have a power of 0.80, alpha = .05, and a small/medium effect size of d = .35.” 

Please clearly state for what test this number of participants is adequate: Is it for an independent samples t-test? or for the mediation analysis? Furthermore, when chosing the alpha level, how was the number of tests considered (e.g., through Bonferroni correction?)

We have now added in this information (p.6, p.12, p.15). Power calculations were conducted for independent sample t-tests (Study 1) and ANOVAs (Studies 2 and 3). We did not correct for multiple comparisons, but instead decided to confirm significant results with preregistered replication studies.

1.5. Minor points: 

• I found the wording “independent T-tests” (e.g., Page 8, “results and discussion” section) a bit confusing. I suggest to write: “independent sample t-tests” (lowercase)

Thanks for this correction - we have changed this now.

• Page 9, also the following sentence was a bit confusing: “model 4 was used to test parallel mediation).” What do you mean exactly? Does it mean that you run two separate analyses, one for each dependent variable?

We have clarified this and state that we ran two separate analyses for each independent variable (p. 9)

“We conducted parallel mediation analyses (Hayes, 2013, model number 4) to test the mediating role of both human superiority beliefs and demoralization of veganism. We tested two mediation models – one with positive attitudes towards meat eating as the outcome variable, and one with positive behavioral intentions towards veg*nism as the outcome variable.”

• Study 1, the mediation analyses would be easier to follow if you could present them graphically, in a similar way to what you did for study 2b

We have now added figures for these mediation models in Study 1 too. 

• I wonder why you chose to name them Study 1 - 2a - 2b, instead of Study 1 - 2 -3, especially because 2a and 2b are not identical to each other, so I found this a bit confusing (but this may be a matter of personal preferences).

On reflection upon reading your comment we agree that there are important differences between these studies. For this reason and to avoid any confusion, we have now labelled studies 1, 2, and 3. 

Reviewer 1

2.1. General feedback

Thank you for including me in this process. I thoroughly enjoyed the paper and think that it is a valuable contribution to the field.

We thank you for your endorsement of our work and appreciate your helpful feedback. 

2.1 Additional references 

The main issue I have with the manuscript is the exclusion of many relevant studies in the introduction, and especially in the discussion. I have suggested a few (there are more) at the end of the review. This literature will help unpack and discuss results in relation to possible contextual features, moralisation/demoralisation, efficacy etc.

Thank you very much for drawing our attention to these important papers. We have now discussed much of this literature in the introduction and discussion, as well as other more recent relevant work, including:

Caviola, L., Everett, J., & Faber, N. S., 2019

Cruwys, Norwood, Chachay, Ntontis, & Sheffield, 2020 

Leite, A. C., Dhont, K., & Hodson, G., 2019 

Rosenfeld, Rothberger & Tomiyama, 2020

Shwarz, O., 2020

We believe these additions have indeed strengthened the discussion and manuscript. 

2.2. Israeli diet context 

I think it would be useful to expand on the discussion about the Israeli diet context you touch upon that there is a need to test this in further contexts (p.27), but it would be useful for the reader to know more about the Israeli context and why there might be contextual differences (or not) – see for example Schwartz, 2020).

We have added in information about the Israeli diet context (please see our response to comment 1.1). The Schwarz reference was very helpful. 

We suggest that Israel shares important contextual similarities with other developed countries, both in light of the high animal-product consumption and the social identities of veg*ns (mainly women, young, and liberal). 

2.3. Discussion on identities 

I was also expecting a discussion on identities/values attached to groups, such as the issues with merging all non-meat eaters even though you are focusing on omnivores.

The context could also feed into the moralisation/demoralisation in terms of group belonging/cultural belonging evoked by the different condition texts – you almost touch upon this on p.27 (“but further research is merited to examine which are the more meaningful defense mechanisms and for whom.”) but I think you could elaborate with a sentence or two.

Thank you for this comment. We have added in a paragraph in the discussion addressing the role that identity and culture may play in people’s responses to different types of pro-veg*nism messages (p.22):

“Defensive responses might be more or less pronounced among those with affinities to different social identities or cultures. Existing work suggests that there are multiple factors that can influence meat-eating behavior, including values, beliefs, emotions, or external incentives such as price and availability, and that these may vary for groups of people with particular identities (Stoll-Kleemann & Schmidt, 2017). In the real world, veg*n animal advocacy groups not only make moral arguments for veg*nism, they also employ other strategies in a bid to reduce animal product consumption, such as promoting personal health messages, promoting flexible diets, and drawing on social norms. It could be that for those who might be more inclined towards veg*nism for ethical reasons at the outset, moral messages, like the ones we tested in our study, may have more influence. However, for other groups of people who are more motivated to preserve their health, such as older populations, health messages may be more effective (Stoll-Kleemann & Schmidt, 2017). This is a question that future work could explore.”

We also acknowledge that being veg*n may be incompatible with certain social and cultural identities, as mentioned by Schwarz (2020).

2.4 Elaborating on Practical implications 

Finally, I am missing a more direct discussion of the practical implications of your research, you have a brief description of what to do and not, but this could be elaborated more on in terms of value for campaigns, social movements etc. (even beyond the veganism) to emphasise the usability.

The key message that we would like activists and campaigners to take from our findings is that people, in general, care deeply about their moral image and can become highly defensive when it is threatened. More than just providing important facts and information, we suggest that this information should be provided in a way that maintains the integrity of their target audience’s moral self-image, in order not to elicit defensive responses. We have now elaborated on this point in the general discussion (p.22):

“From an applied perspective, our results suggest that activists and campaigners ought to seriously consider that people care deeply about their moral image. Attributing blame may be an instinctive response for those who are invested in moral issues (also beyond veg*nism). However, activists would do well to impart powerful and thought-provoking information while maintaining the integrity of their target audience’s moral self-image, in order not to elicit defensive reactions. If information is provided in a way that can be construed as blaming, self-protective and justifying responses may follow and campaigns may even backfire.”

2.5 Some minor (grammatical) points:

Inconsistent use of v*gn and veg*n

(p.20) “Furthermore, those the blaming text was perceived as significantly more blaming” – missing a ‘reading/assigned’

Thank you. We have fixed these typos now. 

Reviewer 2

3.1 Discussion of potential benefits of a blame frame

You do a good job of showing the practical problems with using a blame frame (efficacy issues in having it backfire psychologically resulting in rejection), but I think you should also mention what benefits may come from using a frame that puts some responsibility on the consumers themselves who finance this industry (or them as citizens who elect political leaders that endorse these farming laws and practices). If the blame is warranted, is there something valuable in admitting a consumer’s culpability in the problem while also showcasing how they are then a part of the solution? Or is the ultimate test of whether a campaign is “successful” based on immediate responses to messages and short term behavior change? Might there be long-term or deeper benefits to assigning a blame frame that an experiment cannot measure? (to add to your limitations section). This could be acknowledged.

A question to consider (although it was not tested in the study but could be acknowledged as an area for future research): Could the blame be allocated more collectively/socially (our society/culture is problematic because it validates and socially sanctions this mass consumption of animals) rather than aimed directly at them as one individual consumer? And would this not reduce the defensiveness and perhaps engage people in terms of citizenship (not just their role as consumer).

Thank you very much for raising these fascinating questions. 

First, we now stress in the introduction that absolving messages still involve holding people accountable for their future actions (p.4-5). It was also our assumption that holding people responsible for their harmful behavior is important and may have benefits, but we expected that it would have less benefits if done in a way that is also construed as blaming:

“Absolving messages suggest that omnivores are not fully aware of the entailed cruelty, and therefore cannot be held responsible for it. Importantly, absolving does not justify the harm done to animals nor does it suggest that omnivores are less responsible for their future animal-product consumption; it simply suggests that individuals are less to blame for their past involvement in the harmdoing.”

We agree that it would be very interesting for future work to examine both the role of blame assigned at the collective level and the long-term effects of blame. We now mention both these points in the general discussion (p. 24-25):

“An interesting question that is beyond the scope of this paper is whether assigning blame at the group level (e.g., society, omnivores) would be more effective than blaming the individual for their past harmdoing. While one could argue that the threat associated with blame may be diffused if targeted at a collective, we would expect that because one’s identity and self-esteem is generally strongly intertwined with one’s group identity (Tajfel &Turner, 1979), blame directed at one’s group may be equally counter-productive. Finally, examining the long-term effects of being blamed is important for both theoretical and applied purposes. More time to consider a blaming message may simply provide people with greater opportunity to further justify their opinions and could thus entrench their existing attitudes even more. However, given that our work looked at the immediate response to blaming messages, more research is necessary to test this empirically.” 

3.2. General feedback

METHOD: Decent sample size of 300+ study 1 –recruited via social media (and had more women). Doubled for study 2a bc paid participants from a hired survey company. Good that you noted the differences in sample sourcing may affect results. Study 2b (316 participants, many more women). Plus a mini meta analysis. While I am not a quantitative researcher, from what I could assess, I found the three messages (blaming, absolving, and neutral) to be appropriate as a testing tool, as well as the measures for other categories such as 4Ns, human superiority, dietary change efficacy, etc. (in reading your supplementary materials). I could tell you as researchers understand vegan advocacy and animal rights issues.

I like that in study 2a you considered contravening variables and “therefore, measured efficacy beliefs as an exploratory variable, and considered that reduced efficacy beliefs may be an additional defense mechanism utilized by those who are blamed.” In general, I find that the researchers were self aware in acknowledging weaknesses/limitations and potential confounding variable.

I also like that you ran an extra test with study group one to test some inconsistencies from Study 2a: “Given the inconsistency between the strong effects in Study 1 and the null effects in

Study 2a, we decided to re-run the study with participants recruited via social media like in

Study 1.” (133 participants: many more women).

Thank you very much for these encouraging comments and your thorough review of the manuscript. 

3.3 Applied consequences of the findings regarding the effect of blame on the hypothesized mediators 

I think you should explain further what is meant by this finding in study 2b using terms that activists could understand. “There were several defense mechanisms that mediated these effect namely, demoralization, reduced efficacy beliefs, and reactance, though not human superiority.” What implication does this have as far as providing tangible guidance for campaign designers? 

A question that arose for me after reading study 1: Are you saying that being blamed for farmed animal suffering won’t make you rationalize meat eating as much as it will make you defend your human superiority over other animals? (although either as well as both would be detrimental to dietary behavior change). This would be an area for more granular/nuanced study in what messages may be effective here in seeing if it is human superiority beliefs that most need changing among omnivores rather than focusing on changing beliefs about meat eating. Or perhaps you can think of more specific ways this finding is significant for vegan advocacy campaign designers and could be further examined by scholars.

Thank you for raising these important questions about the mechanism, and the practical implications of our related findings. 

For clarity, we have added this to the discussion of Study 3 (previously named 2b) (p.20):

“In other words, after being blamed (vs. absolved), people regarded veg*nism as less of a moral issue, believed that they were personally less able to change their diet towards veg*nism, and felt that the information that they read about animal suffering (with the blame-frame) was trying to threaten their freedom of choice.”

We agree that understanding when specific defense mechanisms are used over others, and who is more inclined to use particular mechanisms, is an important area for future work, and include this point in the discussion (p. 24).

For applied purposes, we think it would be premature to conclude that activists should develop messages to tackle particular rationalizations over others. Instead, we think that the take home message of our findings is that messages need to be relayed in a way that is not perceived as blaming and does not lead people to put up some kind of defense, as our findings indicate that defensiveness, in general, leads to resistance to change. Activists would do well to understand that people place great value on their moral self-image and that when it is threatened, people will often respond defensively in some way. We have now added this to the general discussion to emphasize our applied recommendations (please see our response to comment 2.4):

3.4. Grammatical points

Make “effect” plural in the middle sentence “There were several defense mechanisms that mediated these effect namely”

Thank you for catching this typo – we have fixed it.

---

## [Editor Report · Decision Letter 1]

25 Jun 2021

The Cost of Attributing Moral Blame to Omnivores when Raising Awareness to Animal Suffering in Factory Farming: Defensiveness and Resistance to Change

PONE-D-20-39949R1

Dear Dr. Reifen-Tagar,

We’re pleased to inform you that your manuscript has been judged scientifically suitable for publication and will be formally accepted for publication once it meets all outstanding technical requirements.

Kind regards,

Cristina Zogmaister, Ph.D.

Academic Editor

PLOS ONE

Additional Editor Comments (optional):

I have read with great pleasure the new version of your article. I believe that you have responded convincingly to the points raised by me and the reviewers and that your work is a useful research. It is therefore with great pleasure that I accept your publication of this research.
---

## [Editor Report · Acceptance letter]

29 Jul 2021

PONE-D-20-39949R1 

The Cost of Attributing Moral Blame: Defensiveness and Resistance to Change when Raising Awareness to Animal Suffering in Factory Farming 

Dear Dr. Reifen-Tagar:

I'm pleased to inform you that your manuscript has been deemed suitable for publication in PLOS ONE. Congratulations! Your manuscript is now with our production department. 

Kind regards, 

on behalf of

Dr. Cristina Zogmaister 

Academic Editor

PLOS ONE